# Silver Nanoparticle Production Mediated by *Vitis vinifera* Cane Extract: Characterization and Antibacterial Activity Evaluation

**DOI:** 10.3390/plants11030443

**Published:** 2022-02-05

**Authors:** Jana Michailidu, Olga Maťátková, Irena Kolouchová, Jan Masák, Alena Čejková

**Affiliations:** Department of Biotechnology, Faculty of Food and Biochemical Technology, UCT Prague, Technická 5 Dejvice, Praha 6, 166 28 Prague, Czech Republic; olga.matatkova@vscht.cz (O.M.); irena.kolouchova@vscht.cz (I.K.); jan.masak@vscht.cz (J.M.); alena.cejkova@vscht.cz (A.Č.)

**Keywords:** nanoparticles, biosynthesis, antimicrobial activity, plant extract

## Abstract

The ever-growing range of possible applications of nanoparticles requires their mass production. However, there are problems resulting from the prevalent methods of nanoparticle production; physico-chemical routes of nanoparticle synthesis are not very environmentally friendly nor cost-effective. Due to this, the scientific community started exploring new methods of nanoparticle assembly with the aid of biological agents. In this study, ethanolic *Vitis vinifera* cane extract combined with silver nitrate was used to produce silver nanoparticles. These were subsequently characterized using UV-visible (UV-Vis) spectrometry, transmission electron microscopy, and dynamic light-scattering analysis. The antimicrobial activity of produced nanoparticles was tested against the planktonic cells of five strains of Gram-negative bacterium *Pseudomonas aeruginosa* (PAO1, ATCC 10145, ATCC 15442, DBM 3081, and DBM 3777). After that, bactericidal activity was assessed using solid medium cultivation. In the end, nanoparticles’ inhibitory effect on adhering cells was analyzed by measuring changes in metabolic activity (3-(4,5-dimethylthiazol-2-yl)-2,5-diphenyltetrazolium bromide assay-MTT). Our results confirmed that ethanolic *Vitis vinifera* cane extract is capable of mediating silver nanoparticle production; synthesis was conducted using 10% of extract and 1 mM of silver nitrate. The silver nanoparticles’ Z-average was 68.2 d nm, and their zeta potential was –30.4 mV. These silver nanoparticles effectively inhibited planktonic cells of all *P. aeruginosa* strains in concentrations less than 5% *v/v* and inhibited biofilm formation in concentrations less than 6% *v/v*. Moreover, minimum bactericidal concentration was observed to be in the range of 10–16% *v/v*. According to the results in this study, the use of wine agriculture waste is an ecological and economical method for the production of silver nanoparticles exhibiting significant antimicrobial properties.

## 1. Introduction

Nanoparticles rose to the forefront of scientific interest decades ago; however, their significance and potential is still being extended by new discoveries. Their unique physical and chemical properties, mainly caused by their high specific surface area, are already being used in a wide range of science and industry, from electronics or material design to food processing [1]. In recent years, though, another characteristic of nanoparticles started to become more and more important. With the rise of resistant pathogenic microorganisms, which cause infections not responding to traditional antibiotic therapy, there appeared a great need to find new ways of combating these once easily curable illnesses. Metal nanoparticles, in particular, have since been extensively studied for their antimicrobial properties against the dreaded resistant pathogenic microorganisms. For example, silver, gold, copper, platinum, and palladium nanoparticles have already proven their inhibiting effect on pathogenic bacteria such as *Pseudomonas aeruginosa, Escherichia coli, Staphylococcus aureus* (including methicillin resistant *Staphylococcus aureus*), *Bacillus subtilis,* and *Streptococcus bovis* [2].

Still, the ever-growing potential of metal nanoparticle applications foreshadows future problems with their mass production. Physico-chemical methods of nanoparticle manufacture are not cost-effective and can have dire environmental consequences due to the use of toxic solvents. The production of nanoparticles using biological agents, however, promises comparable yields and stability without wasting energy and damaging the environment [3]. Biomolecules such as proteins, amino acids, vitamins, reducing sugars, terpenoids, polyphenols, and organic acids act as reducing and stabilizing agents in this type of nanoparticle production. One of the proposed mechanisms in plant extract mediated synthesis involves polyphenols such as flavonoids or quercetin. The hydroxyl groups present in polyphenols reduce metal ions, themselves oxidizing to carbonyl groups. The resulting carbonyl groups subsequently electrostatically stabilize the nanoparticles [4]. In a study by Shameli et al. [5], silver nanoparticles (AgNPs) were synthesized using stem bark extract of *Callicarpa maingayi.* It was identified that aldehydes contained in the extract reduced silver ions to metallic AgNPs, forming complexes with amide and polypeptide groups, preventing the NPs from aggregation. The presence of biologically formed capping agents is not considered a hindrance, especially in the context of the antimicrobial effect; on the contrary, the capping molecules could have a synergistic effect with the action of nanoparticles. Moreover, the capping agents of biological origin could potentially lower the toxicity of nanoparticles for human cells [6].

In this regard, multiple different approaches were already employed and reported in the literature; live microorganisms, microbial lysates, plant tissue, and even plant extracts were successfully used as a source of reducing and stabilizing molecules for nanoparticle production [7,8,9,10,11]. When it comes to the microbial production of nanoparticles, *Bacillus cereus, Escherichia coli,* and *Pseudomonas proteolytica* have proven their ability to form silver nanoparticles, whereas *Shewanella alga* and *Rhodopseudomonas capsulate* were able to produce gold nanoparticles [12,13,14]. In regard to the use of plant extracts, *Acalypha indica, Boswellia serrata, Cinnamon zeylanicum,* and *Withania somnifera* were used for the production of silver nanoparticles [15,16,17,18]. In a study by Gnanajobitha et al. [19], *Vitis vinifera* fruit extract was used for the synthesis of AgNPs. In another study by Saratale et al. [20], tannin extracted from grape pomace was used for the mediation of silver nanoparticle production. A variety of different parts of plants are used in the literature: leaves, flowers, seeds, roots, fruits, stems, and peels [21]. The most industrially feasible solution, however, seems to be the use of extracts made from dry plant material, which can be easily stored. Moreover, this material should preferably not have any other major purpose (e.g., agricultural waste).

In this study, we observed the antimicrobial activity of AgNPs, produced using *V. vinifera* extract made from agricultural waste and silver nitrate solution, against the planktonic cells of *Pseudomonas aeruginosa*.

## 2. Results

### 2.1. Nanoparticle Biosynthesis and Characterization

After the nanoparticle synthesis was completed, UV-visible (UV-Vis) spectrophotometric analysis was carried out, and it revealed a distinct peak at 430 nm (see Figure 1). This peak was apparent with all reagent ratios used (10% *v/v* of *V. vinifera* extract with 0.1, 0.25, 0.5, 1, or 2 mM of silver nitrate; 1 mM of silver nitrate with 0.5, 1, 2.5, 5, 10, 15, or 20% *v/v* of *V. vinifera* extract)—data not shown. The results from this screening experiment were used to choose a reagent ratio producing the highest and narrowest peak (1 mM of silver nitrate and 10% of *V. vinifera* extract); nanoparticles for subsequent antimicrobial experiments were produced accordingly. After that, the stability of produced AgNPs stored in the dark at room temperature was assessed over a period of 2 months (see Figure 1).

After that, AgNPs were subjected to dynamic light-scattering (DLS) analysis (see Table 1). The zeta potential of AgNP suspension was determined as –30.4 mV, where zeta potential values exceeding –25 or 25 mV signify a stable nanoparticle suspension, which will likely not aggregate due to van der Waals interactions [22]. The size of produced AgNPs was characterized using the DLS analysis; the Z-average was calculated as 68.18 d nm. The polydispersity index (PdI) of AgNP suspension was 0.214.

The results of the DLS size analysis were later confirmed by photographs from transmission electron microscopy (TEM) of the AgNPs (see Figure 2). TEM analysis revealed that the produced AgNPs are predominantly spherical in shape, and the particle size distribution of the suspension is relatively high, with most nanoparticles having a diameter of around 13–18 nm (see Figure 3).

The yield of nanoparticle synthesis was assessed using atomic absorption spectrometry (AAS). It was discovered that 65% of Ag ions present in the initial mixture were transformed into nanoparticles, making the final concentration of AgNPs 70.1 mg/L, with 37.8 mg/L of remaining silver ions.

### 2.2. Minimal Inhibition Concentration of AgNPs against Planktonic Cells of P. aeruginosa

The antimicrobial activity of the AgNP suspension was able to inhibit the growth of planktonic cells of all the *P. aeruginosa* strains tested (see Table 2). MIC_80_ was observed at 3.13% *v/v* of AgNPs for *P. aeruginosa* DBM 3081 and DBM 3777. The planktonic cells of *P. aeruginosa* ATCC 15442 were inhibited by more than 90% in the presence of 2.50% *v/v* of AgNPs. Regarding the last two strains, *P. aeruginosa* ATCC 10145 and PAO1, 1.88% *v/v* of AgNPs was needed for total inhibition.

### 2.3. Bactericidal Activity of AgNPs against P. aeruginosa Cells

The bactericidal activity experiment was carried out to find the MBC of biosynthesized AgNPs (see Table 3). The MBC was observed for all the strains tested apart from *P. aeruginosa* ATCC 10145 in the concentration range used. For both *P. aeruginosa* DBM 3081 and *P. aeruginosa* DBM 3777, the MBC was determined as 15.63% *v/v* of AgNPs. Other strains exhibited even lower tolerance for the biosynthesized AgNPs. The MBC against *P. aeruginosa* ATCC 15442 was determined to be 12.50% *v/v* (see Figure 4), and the MBC against *P. aeruginosa* PAO1 was determined as 10.71% *v/v*.

### 2.4. Biological Activity of AgNPs against Adhering Cells of P. aeruginosa

After the effect of biosynthesized AgNPs on planktonic cells was evaluated, the experiments exploring the ability of AgNPs to inhibit cell adhesion were carried out (Table 4). MBIC_50_ was found in the range of 3.21–5.36% *v/v* for all the strains studied apart from *P. aeruginosa* DBM 3777. MBIC_90_ was 3.93, 5.36, and 5.36% *v/v* when assessing the adhesion inhibition against the cells of PAO1, *P. aeruginosa* ATCC 15442, and *P. aeruginosa* DBM 3081, respectively. MBIC_90_ was not found for either *P. aeruginosa* ATCC 10145 and *P. aeruginosa* DBM 3777 in the concentration range tested.

## 3. Discussion

In the age of antibiotic resistance, there is growing pressure to inhibit the growth of pathogenic microorganisms using alternatives to traditional antibiotic agents. Metal nanoparticles have thus become a subject of research in this regard [23]. However, their wider application is being partially hindered by inconveniences regarding the physico-chemical methods of their production. Therefore, “green” approaches to nanoparticle production have been studied intensively in recent decades [24]. In this study, we used agricultural waste, in adherence to circular economy principles, for the mediation of silver nanoparticle production.

AgNPs in this study exhibited a distinct peak at 430 nm when subjected to UV-Vis spectrophotometry. According to the literature, a peak between 410 and 450 nm is characteristic of silver nanoparticle suspension. In a study by Packialakshmi and Naziya [25], a peak at 420 nm was observed when silver nanoparticles were synthesized with the aid of *Garcinia mangostana* stem extract. A similar study by Bharathi et al. [26] was carried out, observing nanoparticle formation after the addition of *Diospyros montana* bark extract to silver nitrate solution, after which the UV-Vis spectra analysis revealed a peak at 432 nm. In a study using tannin extracted from grape pomace, silver nanoparticle synthesis was confirmed by a distinct peak at 420 nm [20]. In another study, using *V. vinifera* fruit extract, it was found that a peak at 450 pointed to a successful synthesis of silver nanoparticles [19]. Using multipole scattering theory, it was found that the position of a UV-Vis spectrophotometric peak can be used to calculate the size of nanoparticles produced, with higher wavelengths implying larger nanoparticles and vice versa [27].

Through the DLS analysis of AgNPs, in this study, we determined the average diameter as 68.18 nm and the PdI to be 0.214. In a study by Nayak et al. [28], the DLS analysis revealed that nanoparticles synthesized using *Ficus benghalensis* and *Azadirachta indica* bark extracts had average diameters of 85.95 and 90.13 nm, respectively, and a PdI of 0.247 and 0.314, respectively. In another study by Rao and Tang [29], it was demonstrated that reaction temperature can have a direct impact on average size and PdI. Their analysis showed that AgNPs synthesized with the aid of *Eriobotrya japonica* leaf extract had an average diameter of 76.10 nm with a PdI of 0.571 at 25 °C and an average diameter of 54.47 nm with a PdI of 0.281 at 80 °C. AgNPs produced in this study exhibited a relatively low PdI; this implies a more uniform nanoparticle suspension, which can be an advantage in the case of medicinal applications.

According to the DLS analysis, AgNPs in this study exhibited a zeta potential of −30.4 mV, which insinuates their long-term stability. In a study by Ravichandran et al. [30], the characterization of AgNPs produced using *Atrocarpus altilis* leaf extract revealed a zeta potential of −12.7 mV. A study by Anandalakshmi et al. [31] demonstrated silver nanoparticles synthesized using *Pedalium murex* leaf extract with a zeta potential of only −7.66 mV. In another study by Singh et al. [32], the zeta potential of silver nanoparticles synthesized with the aid of *Cannabis sativa* fiber extract was −29.2 mV. This value implied the stability of the AgNPs produced, which was subsequently confirmed by repeated UV-Vis spectra assessment (see Figure 1).

TEM analysis showed the morphology of AgNPs produced in this study to be spherical. Moreover, TEM analysis showed the size of AgNPs as being between 1 and 30 nm. In a study by Sadeghi and Gholamhoseinpoor [33], it was discovered by TEM analysis that AgNPs formed using *Ziziphora tenuior* extract were spherical in shape and that their size ranged from 8 to 40 nm. Another study by Ahmed et al. [34] also reported spherical AgNPs with a diameter of 34 nm when *Azadirachta indica* extract was used as the reducing and stabilizing agent. Irregular spherical shape and size between 20 and 100 nm was described in a study by Vijayakumar [35], where *Justicia gendarussa* extract was used for the phytofabrication of AgNPs.

Although AgNPs have been proved to exhibit powerful antimicrobial action against both Gram-negative and Gram-positive bacteria (*Staphylococcus aureus, Bacillus cereus)*, viruses (HIV 1), and fungi (*Candida, Aspergillus*), we will focus on exploring their activity against *P. aeruginosa* [36]. When it comes to planktonic cell inhibition, the dispersion in this study was effective in the range of 1.32–2.21 mg/L of AgNPs with a remaining 0.71–1.18 mg/L of silver ions against all the strains tested. To compare the results with a traditional antibiotic, MIC_80_ of polymyxin B was observed between 2.5 and 3.5 mg/L against *P. aeruginosa* cells [37]. Regarding other studies using AgNPs, in a study by Pompilio et al. [38], the planktonic cells of *P. aeruginosa* were inhibited by AgNPs in the range of concentrations 1.06–4.25 mg/L. In another study by Singh et al. [39], AgNP MIC against multidrug-resistant strains of *P. aeruginosa* was observed to be between 6.25–12.5 mg/L. Finally, in a study by Radzig et al. [40], it was described that MIC of silver ions against *P. aeruginosa* PAO1 is 0.3 mg/L, and MIC of AgNPs is 8.0 mg/L. According to these results, combining silver ions with AgNPs could lower the concentration of silver necessary for the successful inhibition of the planktonic cells of *P. aeruginosa* compared to using silver ions or nanoparticles alone. The AgNP suspensions produced in this study combined the actions of nanoparticles and silver ions and proved more effective than the AgNPs in other cited studies.

In the case of biofilm formation inhibition, MBIC was achieved when 2.27–3.79 mg/L of AgNPs and 1.21–2.02 mg/L of silver ions were used. In a study by Palanisamy et al. [41], a 67% inhibition of biofilm formation of *P. aeruginosa* was achieved when 20 mg/L of AgNPs was used. In a different study by Markowska et al. [42], it was observed that AgNP MBIC against *P. aeruginosa* is 4 mg/L. In comparison, MBIC_80_ of polymyxin B against *P. aeruginosa* was found to be between 5 and 15 mg/L [37]. Regarding the last antimicrobial experiment, total bactericidal effect was observed when 7.57–11.05 mg/L of AgNPs with 4.05–5.91 mg/L of silver ions was used. In a study by Lara et al. [43], the MBC was found to be in the range of 8.88–10.7 mg/L. In another study by Mohan et al. [44], the bactericidal activity against *P. aeruginosa* was achieved when 12.5 mg/L of AgNPs was used. The combined effect of nanoparticles and silver ions in AgNP suspension produced in this study provided at least two-times stronger antibiofilm and bactericidal activity than the nanoparticle suspensions in other studies.

According to the results of experiments in this study, AgNP suspension manufactured using agricultural waste from wine production can be a cheap and effective alternative to traditionally produced nanoparticle suspensions when it comes to bactericidal activity or the inhibition of both the planktonic cells and the formation of biofilm.

## 4. Materials and Methods

### 4.1. Materials for Biosynthesis

*V. vinifera* canes, from both white and red varieties that were obtained during the dormancy period in January 2017 from vineyards in the Czech Republic, silver nitrate (Sigma Aldrich, St. Louis, MO, USA), ethanol (40%), and distilled water.

### 4.2. Extraction Method

The extraction method was modified according to Rollová et al. [45]. Briefly, *V. vinifera* canes were homogenized using a blade blender. Then, 150 g of the homogenized canes were mixed with 600 mL of 40% ethanol solution. This mixture was macerated for 24 h; then, it was filtered through an 8 μm filter paper and then again through a 0.2 μm filter paper. The liquid ethanolic extract was stored at 7 °C for further experimentation. The extract was characterized using polyphenolic content determination, high performance liquid chromatography, and ultra-high performance liquid chromatography coupled with high-resolution mass spectrometry in an earlier study [45].

### 4.3. Green Synthesis of Silver Nanoparticles

For the synthesis of AgNPs, a green synthesis method was employed using ethanolic extract of *V. vinifera* and silver nitrate solution. The synthesis was carried out using a number of different reagent ratios. First, 10 mL of *V. vinifera* ethanolic extract (see above) was added to 90 mL of silver nitrate dissolved in water to obtain various final concentrations (0.1, 0.25, 0.5, 1, 2, and 10 mM of silver nitrate). In the second layout, different amounts of *V. vinifera* ethanolic extract (0.5, 1, 2.5, 5, 10, and 15 mL) were added to silver nitrate dissolved in water at a constant final concentration of 1 mM to make a final volume of 100 mL. The mixture was left to react for 48 h, and the resulting AgNP suspension was then characterized using UV-Vis spectrophotometry. All experiments were performed in eight parallels. The final reagent ratios for antimicrobial testing stock dispersion were picked using these pilot experiments—10% *v/v* of *Vitis vinifera* cane extract and 1 mM of silver nitrate solutions were used (see Figure 1). Physical characteristics (apart from AAS) and antimicrobial properties of AgNPs were observed without separation from *V. vinifera* extract.

### 4.4. Characterization of Nanoparticles

UV-Vis spectrometry was carried out using a Reader Infinite M900Pro (TECAN MTP, Männedorf, Switzerland). Spectral analysis (250–700 nm) results were plotted and evaluated to confirm the presence of nanoparticles. UV-Vis spectrometry was also used for stability assessments when the nanoparticle dispersion spectra were measured 2, 4, 6, and 8 weeks after synthesis, while in the meantime, being stored in the dark at room temperature. After that, the nanoparticles were analyzed by transmission electron microscopy using EFTEM Jeol 2200 FS (Jeol Ltd., Tokyo, Japan). Dynamic light-scattering analysis was also carried out using Malvern Zetasizer Nano ZS (ATA Scientific Pty Ltd., Caringbah, Australia).

The yield of synthesis was assessed using a nanoparticle dispersion filtrate by AAS using Agilent 280FS AA. The nanoparticles were extracted from the dispersion by centrifugation. After that, the concentration of silver in the resulting supernatant was analyzed by AAS. The yield of synthesis was assessed as we compared the concentration of silver in the AgNP dispersion supernatant and the initial concentration of silver ions at the start of the synthesis.

### 4.5. Microbial Strains and Growth Media

These bacterial strains of *P. aeruginosa* were used: PAO1, ATCC 10145, ATCC 15442, DBM 3081, and DBM 3777. Glycerol cryopreserves of the microorganisms were stored at −70 °C.

All *P. aeruginosa* strains were precultivated before each experiment in a Luria-Bertani (LB) liquid medium at 37 °C for 24 h to achieve the exponential phase of growth (100 mL in Erlenmeyer flasks, 150 rpm).

Negative controls were conducted for the antimicrobial activity of the corresponding concentrations of both ethanol and extract to rule out the possibility of interference.

### 4.6. Evaluation of Minimal Inhibitory Concentration (MIC)

The antimicrobial effects of biosynthesized AgNPs against planktonic cells was determined using a microcultivation device Bioscreen C (Growth Curves Ltd., Turku, Finland) by a microdilution method, according to Sharma et al. [46]. The cells of tested *P. aeruginosa* strains were cultivated for 24 h in a microtiter plate in the presence of different% *v/v* of AgNPs. Each experiment was done in 10 parallels. For each well, there was 160 µL of LB medium and 30 µL of inoculum (OD_600_ = 0.1), and the remaining 130 µL were filled with varying ratios of phosphate buffer saline solution and AgNPs (0.31, 0.63, 1.25, 1.88, 2.50, and 3.13% *v/v*). Control parallels were filled with 160 µL of LB medium, 30 µL of inoculum (OD_600_ = 0.1), and 130 µL of phosphate buffer saline solution. After that, the minimal inhibitory concentration (MIC_80_) was determined by evaluating growth curves constructed from the data provided by the microcultivation device as the lowest concentration of antimicrobial agent causing 80% inhibition of growth compared to control after overnight cultivation [47].

### 4.7. Evaluation of Minimal Bactericidal Concentration (MBC)

The bactericidal effect of biosynthesized nanoparticles was determined through a subsequent solid medium (LB agar plates) cultivation of planktonic cells taken from a cultivation in the presence of nanoparticles. The cells of tested *P. aeruginosa* strains were cultivated for 24 h in a microtiter plate in the presence of different% *v/v* of AgNPs derived from their respective MICs (1× MIC, 2× MIC, 5× MIC); each experiment was done in three parallels. Control parallels were filled with 160 µL of LB medium, 30 µL of inoculum (OD_600_ = 0.1), and 130 µL of phosphate buffer saline solution. Subsequently, 10 µL of cell suspension representing each concentration of nanoparticles and the control were inoculated onto a LB agar plate and cultivated for an additional 24 h. After that, the minimal bactericidal concentration (MBC) was determined as a concentration yielding the growth of fewer than five colonies (>99% killing) by evaluating the growth on solid medium, according to Rahal and Simberkoff [48].

### 4.8. Evaluation of Minimal Biofilm Inhibitory Concentration (MBIC)

#### 4.8.1. Cultivation Conditions

The cultivation was carried out in polystyrene 96-well microtiter plates (TPP, Trasadingen, Switzerland), according to Shin and Eom [49]. The cells of *P. aeruginosa* strains were cultivated for 24 h in the presence of different% *v/v* of AgNPs, which were derived from their respective MICs (from 0% *v/v* to 3× MIC). For each well, there was 210 µL of inocculum (OD_600_ = 0.8) with 70 µL of differing ratios of phosphate buffer saline solution and AgNPs. Control parallels were filled with 160 µL of LB medium, 30 µL of inoculum (OD_600_ = 0.8), and 130 µL of phosphate buffer saline solution. Eight parallels were carried out for each concentration of AgNPs, including the control. The microtiter plate was covered with a lid and incubated at 37 °C and 150 rpm for 24 h.

#### 4.8.2. MTT Assay

The metabolic activity of the biofilm was measured using a 3-(4,5-dimethylthiazol-2-yl)-2,5-diphenyltetrazolium bromide (MTT) reduction assay, according to Sabaeifard et al. [50]. The wells were washed three times by saline. Afterwards, 60 µL of glucose solution (57.4 mg/mL), and 50 µL of MTT solution (1 mg/mL) were added into each well. The microtiter plate was then covered with a lid and incubated in the dark at 37 °C and 150 rpm for 1 h. After incubation, 100 µL of wash solution was added to every well, and the plate was put on a shaker at room temperature for 30 min, where the wash solution was a mixture of dimethylformamide, phosphate buffer saline solution, acetic acid, and sodium dodecyl sulfate. After 30 min, color intensity was measured using a Reader Infinite M900Pro (TECAN MTP, Männedorf, Switzerland)at 570 nm. Each experiment was performed in eight parallels. Hence, the minimal biofilm inhibitory concentration (MBIC_50_ and MBIC_90_) was determined as the lowest concentration of nanoparticle suspension which inhibited biofilm formation by 50 and 90%, respectively, compared to control [49].

### 4.9. Statistical Analysis

For the evaluation of microbial growth or bioflm assays data, we used Dixon’s Q test to exclude outlying values. Arithmetic means and standard deviations (SD) were calculated for each concentration tested by mentioned assays in relative percentages (control samples were 100%).

## 5. Conclusions

This study developed a successful method of silver nanoparticle manufacture using a plant extract from agricultural waste. The process of synthesis used in this study is cost-effective and produces stable silver nanoparticles ranging in size between 1 and 30 nm, which can be stored at room temperature for up to two months. Moreover, these nanoparticles exhibit significant inhibiting, bactericidal, and antibiofilm activity against *P. aeruginosa* in concentrations between 2.8 and 15.7% *v/v*. With this simple method, silver nanoparticles for antimicrobial purposes can be synthesized while using agricultural waste, thus adhering to circular economy principles.

## Figures and Tables

**Figure 1 plants-11-00443-f001:**
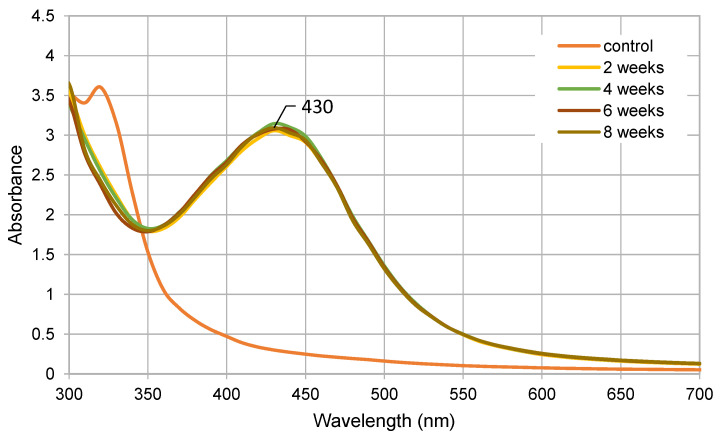
Silver nanoparticle formation using 10% of *V. vinifera* extract and 1 mM of AgNO_3_ and their stability 2, 4, 6, and 8 weeks after synthesis while stored in the dark at room temperature; UV-Vis spectra.

**Figure 2 plants-11-00443-f002:**
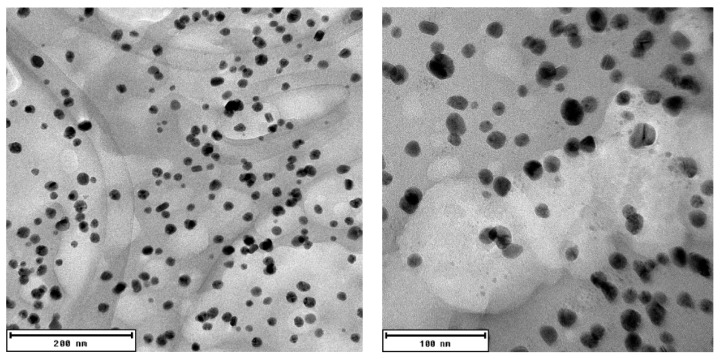
TEM images of AgNPs produced using *V. vinifera* extract.

**Figure 3 plants-11-00443-f003:**
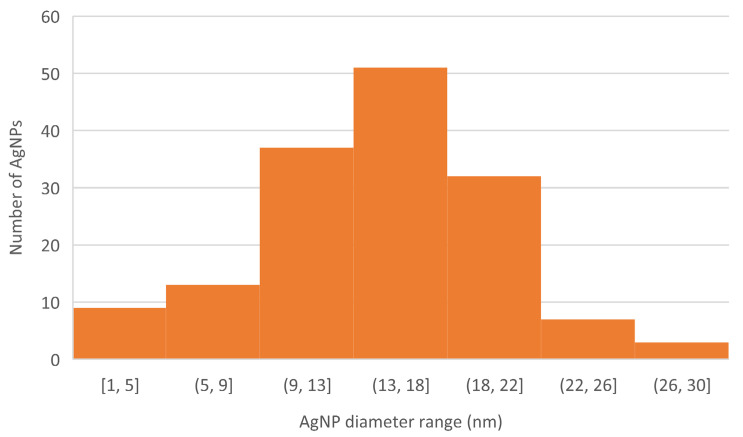
A histogram of TEM image of the AgNPs produced.

**Figure 4 plants-11-00443-f004:**
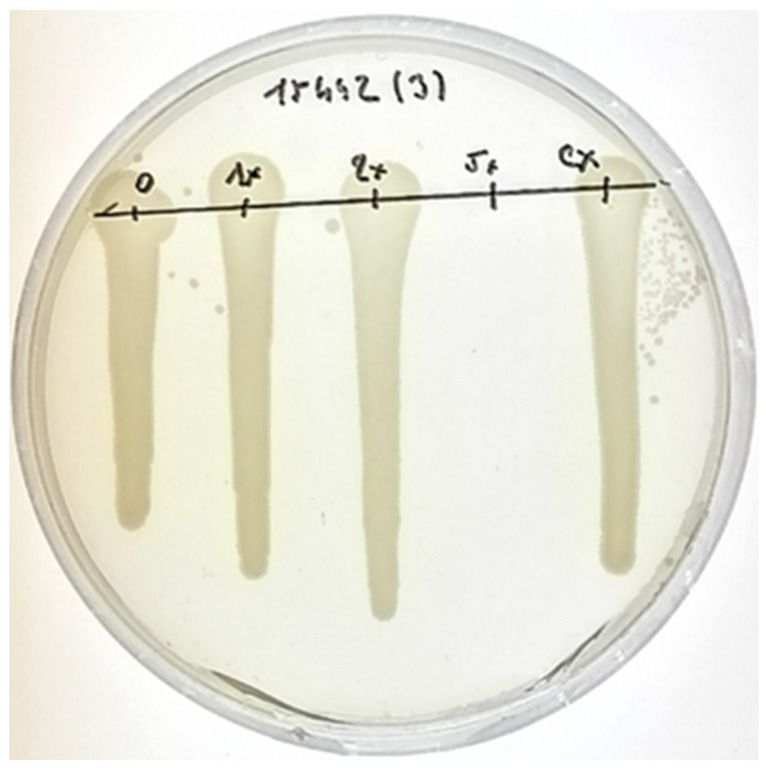
A photograph of a Petri dish with an MBC experiment showing cultivation of *P. aeruginosa* ATCC 15442 without any biological agent (negative control), with different concentrations of AgNPs (1×, 2×, 5× MIC), and a corresponding concentration of *V. vinifera* extract (negative control) (from the left).

**Table 1 plants-11-00443-t001:** DLS analysis results of nanoparticles produced using *V. vinifera* extract.

Zeta Potential (mV)	Z-Average (d nm)	Polydispersity Index
–30.4 ± 3.88	68.18 ± 2.65	0.214

**Table 2 plants-11-00443-t002:** *Pseudomonas aeruginosa* planktonic cell inhibition by AgNPs in *V. vinifera* cane extract; MIC_80_; 100% *v/v* is equivalent to 70.1 mg/L AgNPs with 37.8 mg/L of remaining silver ions.

*Pseudomonas aeruginosa* Strain	PAO1	ATCC 10145	ATCC 15442	DBM 3081	DBM 3777
**MIC_80_ (% *v/v*)**	1.88	1.88	2.50	3.13	3.13

**Table 3 plants-11-00443-t003:** Bactericidal activity of AgNPs in *V. vinifera* cane extract against *P. aeruginosa* strains; 100% *v/v* is equivalent to 70.1 mg/L AgNPs with 37.8 mg/L of remaining silver ions.

*Pseudomonas aeruginosa* Strain	PAO1	ATCC 10145	ATCC 15442	DBM 3081	DBM 3777
**MBC (% *v/v*)**	10.71	-	12.50	15.63	15.63

**Table 4 plants-11-00443-t004:** *Pseudomonas aeruginosa* adhesion inhibition (50 and 80%) by AgNPs in *V. vinifera* cane extract; MBIC_50_ and MBIC_90_; 100% *v/v* is equivalent to 70.1 mg/L AgNPs with 37.8 mg/L of remaining silver ions.

*Pseudomonas aeruginosa* Strain	PAO1	ATCC 10145	ATCC 15442	DBM 3081	DBM 3777
**MBIC_50_ (% *v/v*)**	<3.93	5.00	<5.36	3.21	-
**MBIC_90_ (% *v/v*)**	3.93	-	5.36	5.36	-

## Data Availability

Not applicable.

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
