# Peer review of "Silver Nanoparticle Production Mediated by Vitis vinifera Cane Extract: Characterization and Antibacterial Activity Evaluation"

_plants, 2022, doi:10.3390/plants11030443_

Round 1

Reviewer 1 Report

Presented manuscript is interesting and obtained results may have significant practical meaning. However, some parts need to be improved. Details I have presented below:

  1. In the Introduction part I recommend to add, at least brief information, explaining why natural products are helpful in producing nanoparticles. Using which mechanisms or what kind of chemical reactions.
  2. lines 228-237 - this part must be improved. "10 % v/v of Vitis vinifera extract" - this dilution was prepared using which solvent? (water, ethanol?). "silver nitrate solution" - prepared in water? It must be clearly indicated. "AgNP suspension was then characterized using UV-Vis spectrophotometry" - for what reason it was analyzed uisng UV-Vis and why before the second layout? It must be explained. 
  3. line 241 - "a TECAN Reader" - please indicate producer, city and country
  4. Figure 1 is almost impossible to read. It should be prepared in color.
  5. There is nothing on the repetitions and number of samples used in particular experiments. How can we be sure that the obtained results are repeatable and the method is reproducible?
  6. Why biological acitivity of AgNPs against  P.aeruginosa was not confronted with widely known chemotherapeutic agents, to evaluate their actual activity, and the comparisons were made only with literature data? I understand that the main aim of the Authors was to evaluate the activity of AgNPs against various strains of bacteria, but the results obtained in this way are difficult to assess in terms of their actual activity and practical usefulness.
  7. In the Abstract it must be clearly stated what concentration of Vitis vinifera extract was proved to be the most effective. 

Reviewer 2 Report

Although the manuscript "Silver nanoparticle production mediated by Vitis vinifera cane extract: Characterization and antibacterial activity evaluation" is interesting and has considerable applications in the agronomic sector, the manuscript leaves me somewhat perplexed, especially due to the absence of statistical analysis of the experiments performed (Major point ). Minor points. Fig. 1 is not clear. I advise the authors to better differentiate the experimental lines. The experiment indicated by the authors as data not shown (line 77) must be included in the work. What is the effect of AgNPs on other non-pathogenic and Gram positive bacterial strains? Report at least two experiments in which the inhibition zones (mm) of AgNPs against Pseudomonas aeruginosa are highlighted.

Reviewer 3 Report

Authors present a work in which silver nanoparticles (AgNPs) were synthesised by mixing Vitis vinifera extract with silver nitrate. The physical-chemical characteristics of AgNPs are shown, authors show an improvement of size (smaller) and polydispersity index as compared to other studies. Good antibacterial activity is also shown for these particles. The study is interesting to several fields, shows novelty, however some points need to be clarified and improved before its publication.

Next I will present the several comments/suggestions/questions to this manuscript:

Line 5, please include the country in address

Line 11-12, in my opinion authors should add “silver nitrate” to the sentence as this is the main substrate to produce the AgNPs.

Line 15, please write Gram with a capital letter, as Gram-staining comes from the name of Hans Gram.

Line 16 to 19, please rewrite, the information is not clear, the text needs continuity.

Line 52, replace “potenitally” by “potentially”

Line 79, 85, italicize “Vitis vinifera”

Line 103, define AAS.

Figure 1. Lines should be in different colours or styles to better distinguish them. Please change.

Line 89 – 91, and Table 1. Is the zeta potential negative, or not? As there is a space between the minus symbol and the value, it is not clear if it is a hyphen (as placed in line 234) or if it is a minus sign. This must be clarified. If the values are negative, remove the space between the minus symbol and the value. Hyphens before values should be avoided.

Table 2. Authors present the MIC80 values in % v/v. It will give a better idea about the antimicrobial activity of these NPs if the MIC80 values are given in mg/L of AgNPs. Authors could add a line in table 2 with MIC80 values in mg/L. Or a note as footnote, indicating the initial concentration of NPs and of free silver ions, then it would be easy to calculate the concentration of AgNPs. (This comment also applies to tables 3 and 4.)

Later, when discussing their data (line 187) authors present the data in mg/L; but from tables and from methods the information is not clear, also it is not clear the initial concentration of AgNPs.

It is not clear if dilutions are made from the NPs suspension at 70.1 mg/L (concentration mentioned in line 105), or at other concentration. Methods (lines 264-300) only refer to the dilution factors, do not mention the concentration of the initial NPs solution/suspension. Please add the concentration of AgNPs prior to dilutions. This is an important information.

Tables should be numbered consecutively. The numbering of table 3 and 4 are swapped. Please, revise the text accordingly.

It was not clear to me if the AgNPs used in antimicrobial studies were separated from the Vitis vinifera extract or not. Were the applied AgNPs separated from the producing media? If not which is the antimicrobial effect of the Vitis vinifera extract by itself? Authors mention that to calculate the yield of production the particles were separated by centrifugation, but nothing is mentioned about the NPs used in the following experiments. Please clarify these points.

I would like to know, in the authors' opinion, what is the importance of the Vitis vinifera extract for the production of these NPs that other extracts do not have. Any particular extract component? Do you know the composition of this extracts, or the main chemical characteristics of the extract? For example, total phenols content or the antioxidant activity; were these parameters determined? Or is it the production method itself that is responsible for getting better physicochemical properties of NPs (size, polidispersity, zeta potential)? Some of these explanations would help to enrich the final manuscript.

In Methods section (section 4) authors should number the subsections.

Please read the entire manuscript and correct other typo and grammatical errors.

Reviewer 4 Report

The manuscript presented for review describes the phytosynthesis and application of AgNPs obtained using V. vinifera extracts. Although the literature abounds in studies regarding NPs phytosynthesis, any new work is welcomed, as it provides new knowledge and insights.

However, the present manuscript needs an extensive revision before it can be published. My main observations are:

  1. The introduction needs to be focused more on phytosynthesized AgNPs, epecially those obtained by V. vinifera extracts
  2. Please use the full binomial name of V. vinifera at first apparition in text
  3. The position of the Vis peak is an indicator regarding the NPs dimensions. Please enhance the discussion.
  4. The DLS measurements reveal the hydrodynamic diameter. The diameter of the NPs should also be evaluated from the TEM images. Perhaps a histogram of NPs dimensions should be inserted
  5.  Except UV-Vis spectrometry (with results not sufficiently discussed), I do not find any evidence of AgNPs synthesis (and not, i.e., silver oxide NPs). The authors should have carried out XRD determinations. 
  6. What controls (positive/negative) were used for antimicrobial assays? In my opinion, the negative control should have been an ethanolic solution (if the NPs were not separated, I did not find any evidence regarding the NPs separation) and the positive control a known antimicrobial agent.
  7. The discussion chapter should be enhanced and the results discussed by comparison with literature data regarding AgNPs phytosynthesized using V. vinifera
  8. Conclusions chapter needs re-writing, enhancing on the results obtained
  9. The references list should be enhanced with more recent studies

Round 2

Reviewer 1 Report

Unfortunately, the file with Authors reply was not attached. However, I followed the manuscript and checked how the paper was improved.

In general, the manuscript was improved. However, im my opinion, important parts of the paper still need improvements. Details I have pointed below:

line 51 - this sentence does not sound good. I would rather changed it to: "hydroxyl groups present in polyphenols reduce metal ions, themselves oxidizing to carbonyl groups, and these in turn electrostatically stabilize the NPs"

line 278 - TECAN 278 MTP - plese provide the country of the producer

lines 114-115 - AAS is rather Spectrometry, not spectroscopy

line 263 - once again I ask - what does it mean "10 % v/v of V.itis vinifera extract". From the description in lines 252-258 I guess that fluid extract was obtained and used during synthesis of AgNP. If so, what does it mean 10% v/v of V. vinifera? Fluid extarct diluted in some solvent? In water? Ethanol? Pleae improve this fragment as it is crucial for the whole manuscript.

Author Response

Dear reviewer,

We thank you for your suggestions and we will address them both in the manuscript and in the reply below.

Unfortunately, the file with Authors reply was not attached. However, I followed the manuscript and checked how the paper was improved.

I am sorry to hear that the Authors reply did not reach you. It might have been an error on our side, so I am appending it to the end of this reply.

In general, the manuscript was improved. However, im my opinion, important parts of the paper still need improvements. Details I have pointed below:

line 51 - this sentence does not sound good. I would rather changed it to: "hydroxyl groups present in polyphenols reduce metal ions, themselves oxidizing to carbonyl groups, and these in turn electrostatically stabilize the NPs"

We thank the reviewer for their suggestion, the formulation has been corrected accordingly.

line 278 - TECAN 278 MTP - plese provide the country of the producer

We thank the reviewer for their suggestion, the country has been added.

lines 114-115 - AAS is rather Spectrometry, not spectroscopy

We thank the reviewer for their suggestion, the definition has been corrected.

line 263 - once again I ask - what does it mean "10 % v/v of V.itis vinifera extract". From the description in lines 252-258 I guess that fluid extract was obtained and used during synthesis of AgNP. If so, what does it mean 10% v/v of V. vinifera? Fluid extarct diluted in some solvent? In water? Ethanol? Pleae improve this fragment as it is crucial for the whole manuscript.

We thank the reviewer for their suggestion, the methodology has been supplemented with even more information on the character of the extract.

Here we add the previous Authors reply:

Dear Reviewer,

we are sending the revised manuscript with the changes tracked according to the guidelines given in the previous email from the Editorial Office. We believe we have sufficiently answered all the questions and supplemented all the parts of the manuscript pointed out to be enhanced or clarified.

We thank the reviewer for all their suggestions, and we are looking forward to future cooperation,

Kind regards,

Michailidu Jana

We have added all the suggestions and our answers below:

REVIEWER 1

Presented manuscript is interesting and obtained results may have significant practical meaning. However, some parts need to be improved. Details I have presented below:

  1. In the Introduction part I recommend to add, at least brief information, explaining why natural products are helpful in producing nanoparticles. Using which mechanisms or what kind of chemical reactions.

We thank the reviewer for their suggestion, concrete examples of NP production mechanism have been added to the introduction.

  1. lines 228-237 - this part must be improved. "10 % v/v of Vitis vinifera extract" - this dilution was prepared using which solvent? (water, ethanol?). "silver nitrate solution" - prepared in water? It must be clearly indicated. "AgNP suspension was then characterized using UV-Vis spectrophotometry" - for what reason it was analyzed uisng UV-Vis and why before the second layout? It must be explained. 

We thank the reviewer for their suggestion regarding methodology of biosynthesis. We have clarified the process and added the requested information.

  1. line 241 - "a TECAN Reader" - please indicate producer, city and country

We thank the reviewer for their suggestion, we have added additional information about the device and producer.

  1. Figure 1 is almost impossible to read. It should be prepared in color.

We thank the reviewer for the suggestion, the figure has been colorized.

  1. There is nothing on the repetitions and number of samples used in particular experiments. How can we be sure that the obtained results are repeatable and the method is reproducible?

We thank the reviewer for their suggestion, the number of parallels has been added to the methodology of biosynthesis.

  1. Why biological acitivity of AgNPs against  P.aeruginosa was not confronted with widely known chemotherapeutic agents, to evaluate their actual activity, and the comparisons were made only with literature data? I understand that the main aim of the Authors was to evaluate the activity of AgNPs against various strains of bacteria, but the results obtained in this way are difficult to assess in terms of their actual activity and practical usefulness.

We thank the reviewer for their suggestion. In regards to the positive control, a reference containing information about the action of polymyxin has been added to the discussion.

  1. In the Abstract it must be clearly stated what concentration of Vitis vinifera extract was proved to be the most effective.

We thank the reviewer for their suggestion. Reaction conditions for nanoparticle synthesis have been added to the abstract. 

Reviewer 2 Report

I thank the authors for answering my questions

Author Response

We thank the reviewer for their suggestions.

Reviewer 3 Report

Dear authors

Thank you for answering all the questions.

The corrections/adds made to the manuscript clearly improved its quality.

I am in favour of its publication 

Author Response

We thank the reviewer for their suggestions and for their final evaluation.

Reviewer 4 Report

The authors sufficiently improved the manuscript. It can now be accepted

Author Response

(The authors gave the same response as above.)

Round 3

Reviewer 1 Report

The Authors responded to almost all my questions and have corrected the manuscript accordingly. Howeever, one detail still need to be corrected.

The Authors wrote that they obtained the extract using 40% ethanol and later that "The liquid ethanolic extract was stored at 7° C for further experimentation" (line 256). It is fine. However, later they wrote that "First, 10 % v/v of V.itis vinifera ethanolic extract were  added to silver nitrate" (lines 263-264). So how they prepared ethanolic extract in the concentration of 10%? The crude extract was evaporated and the residue was dissolved to a final concentration of 10%? If so the concentration was m/v and not v/v.

Concentration in v/v may have been obtained only if the fluid extract was dissolved once again with ethanol to a final concentration of 10% v/v. But it does not have more sense to me. But even if it was like that, it must be clearly described, if the method is to be repeatable.

Author Response

Dear reviewer,

we are sending the revised manuscript with the changes tracked according to the guidelines given in the previous email from the Editorial Office. We believe we have sufficiently answered all the questions and supplemented all the parts of the manuscript pointed out to be enhanced or clarified.

We have added all the suggestions and our answers below:

> The Authors responded to almost all my questions and have corrected the manuscript accordingly. Howeever, one detail still need to be corrected.

> The Authors wrote that they obtained the extract using 40% ethanol and later that "The liquid ethanolic extract was stored at 7° C for further experimentation" (line 256). It is fine. However, later they wrote that "First, 10 % v/v of V.itis vinifera ethanolic extract were  added to silver nitrate" (lines 263-264). So how they prepared ethanolic extract in the concentration of 10%? The crude extract was evaporated and the residue was dissolved to a final concentration of 10%? If so the concentration was m/v and not v/v.

> Concentration in v/v may have been obtained only if the fluid extract was dissolved once again with ethanol to a final concentration of 10% v/v. But it does not have more sense to me. But even if it was like that, it must be clearly described, if the method is to be repeatable.

We thank the reviewer for their suggestion. There was a misunderstanding resulting from our description of the preparation of reaction mixture for the synthesis of nanoparticles. In the part of the text where “10 % v/v of Vitis vinifera extract” is mentioned, it means 10 % v/v of extract was added to the reaction mixture. It does not inform the reader about the percentage of ethanol in the extract or in the reaction mixture. The procedure description was clarified to mitigate further misunderstandings by including volumes of the reactants.

Kind regards,

Michailidu Jana